# Location of Firms and Outsourcing

Stefano Colombo [1,*]  and Arijit Mukherjee [2] 

1   Department of Economics and Finance, Università Cattolica del Sacro Cuore, Largo Gemelli 1,
    I-20123 Milan, Italy
2   Industrial Economics Department, University of Nottingham, Nottingham NG8 1BB, UK;
    arijit.mukherjee@nottingham.ac.uk
*   Correspondence: stefano.colombo@unicatt.it

**Abstract:** We analyze the location of final goods producers under spatial competition with strategic input price determination by firm-specific input suppliers when the final goods producers undertake complete outsourcing or bi-sourcing. Under complete outsourcing, the final goods producers locate closer as the distance between the input suppliers decreases, but the distance between the final goods producers may increase or decrease with the transportation costs of the consumers and the transportation costs between the input suppliers and the final goods producers depending on the distance between the input suppliers. The possibility of bi-sourcing reduces the benefit from saving the transportation costs between the input suppliers and the final goods producers, and creates effects which are opposite to those under complete outsourcing. Thus, our results differ significantly from the extant literature considering either no strategic input price determination or strategic input price determination under competition in the input market. We also discuss the implications on the profits, consumer surplus and welfare, and the implications of endogenous location choice of the input suppliers.

**Keywords:** location; input suppliers; outsourcing; bi-sourcing

**JEL Classification:** D43; L11; L13; R32



## 1. Introduction

The Hotelling model (Hotelling, 1929) [1] is the most widely used model to analyze oligopolistic competition between firms selling differentiated products. The underlying idea of the Hotelling model is very simple. The consumers are distributed along a linear segment that can represent either a physical space or a product space. Two firms are located on this linear segment. Each consumer goes to one of the firms and picks up the item which is produced and/or stored there. The firms compete both in the (physical or non-physical) location and in the price. Hence, this class of games is referred to as competitive location models (Kress and Pesch, 2012) [2].

Location-price models have been extensively studied in game theory. Although the seminal contribution of Hotelling (1929) [1] argues that firms tend to agglomerate, the absence of appropriate game theory concepts at that time prevented Hotelling from identifying the correct solution. It is only with d'Aspremont et al. (1979) [3] that it is properly shown; under linear transportation costs of consumers—as assumed by Hotelling—no subgame Nash equilibrium (SPNE) exists for the second-stage equilibrium prices, due to the lack of convexity in consumers' preferences. Hence, d'Aspremont et al. (1979) [3] propose a different scenario, where linear transportation costs are substituted by quadratic transportation costs. In this case, there exists a unique SPNE characterized by maximal differentiation of firms (principle of maximal differentiation), that is the two firms locate at the endpoints of the segment.

Following d'Aspremont et al. (1979) [3], game theory has been extensively adopted to characterize the equilibrium, if any, in terms of location and price when firms compete

in a Hotelling set-up. For instance, Graitson (1980) [4] assumes a max-min strategy for two firms, and he shows that less-than-maximal differentiation is possible in equilibrium. Although these papers assume pure strategies, mixed strategies are possible as well. A first attempt to include mixed strategies into the analysis of the location-then-price game can be found in Osborne and Pitchik (1987) [5], which, however, after showing the existence of an equilibrium in mixed strategies, need to adopt numerical simulations to characterize it. In other cases, scholars adopt SPNE and pure strategies as in d'Aspremont et al. (1979) [3], but extend the Hotelling model by allowing a non-uniform distribution of consumers (i.e., Neven, 1986) [6], more complex spatial frameworks (i.e., Takahashi and De Palma, 1993) [7], or more than two firms (Brenner, 2005) [8]. More recently, the literature on Hotelling has also combined game theory and agent-based modelling to solve the original Hotelling problem (Van Leeuwen and Lijesen, 2016) [9].[1]

Although the extant literature examining the optimal location choice of the final goods producers provides several important insights, the vast literature on the Hotelling model has rarely considered a very simple, but relevant, question of strategic input price determination. In this paper, we use the Hotelling model to examine the implications of input procurement for the final goods producers' profits and their location-price decision when the final goods producers purchase inputs from firm-specific input suppliers. In this respect, we show the implications of complete outsourcing and bi-sourcing (where the final goods producers make inputs in-house and also purchase them from outside input suppliers).[2]

We show under complete outsourcing that the final goods producers locate closer when the input suppliers are closer in space, implying that there is a positive relationship between the location of the input suppliers and that of the final goods producers. Hence, the maximal differentiation result of d'Aspremont et al. (1979) [3] does not occur when the input suppliers are close in space.

When looking at consumer surplus, we find that the consumer surplus decreases with the transportation costs of the consumers, but it increases (decreases) with the transportation costs between the final goods producers and the suppliers if the suppliers are relatively close (far away). The result about the effects of the transportation costs of the consumers is intuitive. If the consumers incur higher transportation costs for purchasing the products, it will reduce consumer surplus. However, the consumer surplus raising effect of higher transportation costs between the final goods producers and the input supplier may sound counterintuitive. It happens for the following reason. On the one hand, higher transportation costs between the firms and the input supplier tend to reduce consumer surplus by raising the prices of the final goods. On the other hand, if the suppliers are close (far away), higher transportation costs between the final goods producers and the input supplier reduce (increase) distance between the final goods producers, which tend to increase (decrease) consumer surplus by reducing (increasing) prices of the final goods. Hence, if the suppliers are far away, both the effects help to reduce consumer surplus for higher transportation costs between the final goods producers and the input supplier. But if the suppliers are close, the second effect can dominate the first effect to increase consumer surplus for higher transportation costs between the final goods producers and the input supplier.

When looking at the effects on welfare, we find that the effects of both types of transportation costs on welfare are similar to those on the consumers. However, the effects of the distance between the input suppliers on welfare are different from those on the consumers. If the input suppliers locate far away, as the distance between them decreases, welfare increases. However, when the distance between them reduces to a certain length, if the distance between the input suppliers reduces further, it reduces welfare. Hence, the distance between the input suppliers and the welfare shows an inverted U-shaped relationship, which happens for the following reason. On the one hand, as the distance between the input suppliers decreases, it reduces the distance between the final goods producers, which tends to increase welfare. On the other hand, as the distance between

the input suppliers decreases, it reduces (increases) the total transportation costs of the consumers for purchasing the final goods and of the final goods producers for purchasing the inputs when the input suppliers are far away (close), which tends to increase (decrease) welfare. Hence, if the input suppliers are far away, both the effects help to increase welfare following a lower distance between the input suppliers. But if the input suppliers are close and the distance between them reduces further, the second effect can dominate the first effect to reduce welfare.

Our paper is related to Matsushima (2004) [18] and Brekke and Straume (2004) [19]. Matsushima (2004) [18] assumes that there are two final goods producers (or firms henceforth) and two input suppliers which are located within a traditional Hotelling segment. The firms must buy all the inputs from the suppliers to produce the final goods. That is, there is complete outsourcing. The suppliers compete in prices and the firms purchase from the supplier with a lower input price. In this framework, he shows that the maximal differentiation result of d'Aspremont et al. (1979) [3] may not occur. If there are large transportation costs between the input suppliers and the firms, as these transportation costs increase, the firms locate closer in space. He also shows that there is an inverse relationship between the location of the suppliers and the firms. If the suppliers locate near the centre, the firms distance themselves maximally, but if the suppliers locate at the opposite ends of the segment, the firms agglomerate.

Differently from Matsushima (2004) [18] and closer to our paper, Brekke and Straume (2004) [19] assume that there is competition between firms but no direct competition between the suppliers, as they consider "bilateral duopolies", or "firm-specific input suppliers". In other words, each firm purchases inputs from firm-specific suppliers, rather than purchasing from markets where the input prices are determined by competition among the suppliers. This is realistic in many situations: asset specificity,[3] by generating "lock-in" effects, inducing the insurgence of "bilateral monopolies" or "bilateral oligopolies", where each downstream firm purchases inputs from a specific supplier, and each supplier can only serve a specific downstream firm (Brekke and Straume, 2004 [19]). Within this framework, Brekke and Straume (2004) [19] mainly consider the implications of bargaining between the firms and the suppliers for the firms' location choice. In particular, they find that the presence of suppliers induces the downstream firms to locate further apart relative to the case where the input suppliers are not considered, and this impact is stronger when the bargaining power of the suppliers is high.

As in Brekke and Straume (2004) [19] and differently from Matsushima (2004) [18], we assume firm-specific suppliers. While assuming that all the bargaining power belongs to the input suppliers, we extend Brekke and Straume's (2004) [19] framework in two important directions.

First, unlike Brekke and Straume (2004) [19], we *do not restrict the suppliers to being located in the same place as the final goods producers*. Hence, unlike them, we consider transportation costs between the suppliers and the final goods producers. We show that the locations of the suppliers play a crucial role in determining the equilibrium outcome by saving the transportation costs between the suppliers and the firms (i.e., creating a cost-saving effect). In this respect, we show the implications of two situations—where the suppliers' locations are exogenous (Section 3), and where they are endogenous (Section 5).

We show that there is a positive relationship between the distance among the suppliers and the distance between the firms, which is opposite to Matsushima (2004) [18]. As a consequence, the equilibrium prices and profits of the firms and the suppliers are positively related to the distance between the suppliers.

In the case of exogenous location of the suppliers, which may reflect the idea that the suppliers also serve firms that are active in other markets and that their location choices reflect this wider customership, the maximal differentiation result of d'Aspremont et al. (1979) [3] may not occur, as in Matsushima (2004) [18]. When the location of the suppliers is endogenous, which happens if the locations of the suppliers are determined by the profits generated in this market, maximum differentiation comes back. Hence, our analysis shows

that whether the maximum differentiation result of d'Aspremont et al. (1979) [3] remains in the presence of strategic input price determination under firm-specific input suppliers may depend on the location choice of the suppliers.

We also show that removing locational constraints for the suppliers and the firms in the spirit of Lambertini (1994, 1997) [21,22] and Tabuchi and Thisse (1995) [23] might alter the impact of the transportation costs on the equilibrium distance of firms. We find that the transportation costs of the consumers might have a negative impact, whereas those between the firms and the suppliers might have a positive impact, which is impossible in Matsushima (2004) [18], where suppliers and firms are constrained to being located within the same segment. These differences are driven by the different input market structure considered in this paper and in Matsushima (2004) [18].

In our analysis with firm-specific input suppliers, each firm has the incentive to locate closer to its supplier in order to minimize the transportation costs ("cost-saving effect"). In contrast, in Matsushima (2004) [18], with competition between the suppliers, each firm chooses the location in order to foster price competition between the suppliers. On the other hand, there is no cost-saving effect at work in Brekke and Straume (2004) [19] because, in that paper, the suppliers and firms are exogenously located in the same place.

Second, we depart from Matsushima (2004) [18] and Brekke and Straume (and the subsequent literature) by incorporating the possibility of bi-sourcing [19], where firms produce part of the inputs in-house and purchase the rest from the suppliers. Although firms often use this make-and-buy strategy, it is ignored in the literature.[4] Hence, we also contribute to the literature by extending the strategy space of the firms through bi-sourcing. As we show, bi-sourcing, which reduces the importance of purchasing from outside suppliers might increase the profits of the firms and the distance between them.

Adding bi-sourcing to our framework yields the following main results. First, if the transportation costs between the firms and the suppliers increase, the distance between the firms increases in equilibrium (unless the suppliers are far apart), which contrasts with Matsushima (2004) [18] and Brekke and Straume (2004) [19] and the results under complete outsourcing. This is due to the fact that when procuring part of the inputs in-house, the cost-saving effect is reduced, so that the incentive for the firms to locate closer to the suppliers reduces. Second, the impacts of the suppliers' location and the transportation costs of the consumers on the equilibrium distance of the firms sharply contrast with the case of complete outsourcing. Thus, bi-sourcing creates important implications for the equilibrium location of firms. Finally, bi-sourcing enlarges the equilibrium profits compared to complete outsourcing when suppliers are sufficiently far apart and the transportation costs between the firms and the suppliers (of the consumers) are high (low) enough.

Besides Matsushima (2004) [18] and Brekke and Straume (2004) [19], there are few other papers considering firms' preference for location in a Hotelling model with strategic input price determination, with and without bi-sourcing. Wang et al. (2019) [25] extend Brekke and Straume (2004) [19] by capturing the intensity of rivalry between the firms, and by allowing the suppliers to use two-part tariffs. They find that maximum distance might arise in equilibrium, depending on the intensity of rivalry, but agglomeration never arises. Moreover, once two-part tariffs are allowed, bargaining power plays no role for the equilibrium distance. However, as in Brekke and Straume (2004) [19], Wang et al. (2019) [25] assume that the suppliers are located in the same place as the firms, so that there is no shipping of the inputs from the suppliers to the firms.[5] Hence, the transportation costs between the firms and the suppliers play no role. Furthermore, they did not consider strategic bi-sourcing.

Matsushima (2009) [26], building on Matsushima (2004) [18], considers the possibility that the suppliers and firms merge, and investigates the implications for the location choices of both the suppliers and firms. In particular, he shows that vertical integration tends to increase firms' distance.[6] Li and Shuai (2017) [29] extend Matsushima (2009) [26] by allowing the suppliers to set two-part tariffs. They show that the manufacturers gain lower profits under vertical separation, and consumers are better off. Li and Shuai (2018) [30] include

R&D investments by the suppliers and investigate the impacts of locational constraints on the equilibrium prices and profits, both under vertical integration and under vertical separation. They find that locational constraints allow the suppliers to better exploit the downstream market and obtain greater profits. Unlike these papers, our purpose is to show the implications of the distance between the input suppliers and the transaction costs between the firms and the suppliers and of the consumers on firms' equilibrium distance in the presence of firm-specific input suppliers and bi-sourcing.[7]

There is a growing body of literature discussing the implications of bi-sourcing, as well as the conditions that make bi-sourcing more likely to arise. For example, Beladi and Mukherjee (2012) [31] examine the incentives for bi-sourcing by introducing competition in the downstream stage and find that higher downstream competition reduces the extent of bi-sourcing in equilibrium. Du et al. (2006) [32] focus on bargaining between a headquarters and internal and external input suppliers and find the conditions related to the timing of the bargaining process that determine the profitability of bi-sourcing compared to complete outsourcing. Colombo and Scrimitore (2018) [33] discuss the role of delegation to managers in Beladi and Mukherjee (2012) [31] and show that it might be profitable for the firms to partially outsource to input suppliers which are less efficient than the final producers, as long as this allows exploiting market advantages induced by delegation.

None of the above-mentioned papers either incorporated a spatial dimension or considered endogenous location of firms. Outsourcing and bi-sourcing in the context of spatial competition have rarely been investigated. The only paper we are aware of is by Lin et al. (2016) [24]. They consider an industry with two downstream firms—one of them can produce the input itself, whereas the other has to purchase it from an external supplier. In addition, the former firm might decide to partially outsource the input production to the non-integrated firm. In such a situation, it is shown that agglomeration between downstream firms might emerge in equilibrium. We differ from that paper in many important ways. First, we impose no asymmetry between the downstream firms, so that both firms have the option to bi-source. Second, we consider firm-specific suppliers rather than a monopolistic supplier. Third, the suppliers can be located far from the firms. Hence, transportation costs between the suppliers and the firms are important for our analysis. Fourth, we also consider the location choice of the suppliers, and impose no restriction on the location choice of the suppliers and the firms.[8,9]

The rest of the paper proceeds as follows. In Section 2, we outline the model. In Section 3, we characterize the equilibrium outcomes under complete outsourcing. In Section 4, we introduce the possibility of bi-sourcing by the firms. In Section 5, some extensions of the model are discussed. Section 6 concludes.

## 2. The Model

We consider a Hotelling model with quadratic transportation costs. There are two final goods producers, Firm $A$ and Firm $B$, which are located within or outside a linear segment of length 1. In other words, as in Brekke and Straume (2004) [19] and differently from Matsushima (2004) [18], the firms are not restricted to being located within the linear segment of length 1. We indicate by $x_J$, $J = A, B$, the location of Firm $J$, with $x_A \leq x_B$, whereas we define $x \in [0, 1]$ as the location of a generic consumer in the segment. The consumers are uniformly distributed along the linear segment of length 1. Each consumer buys one unit of the product from one of the firms and, in doing so, bears quadratic transportation costs as in d'Aspremont et al. (1979) [3]. Therefore, the utility of a consumer buying from Firm $J$ is $v - p_J - t(x_J - x)^2$, where $p_J$ is the price charged by Firm $J$. Parameter $v$ is the reservation price, which is assumed to be positive and sufficiently high to allow for full market coverage, and $t > 0$ is the transportation cost parameter for the consumers.

We follow Matsushima (2004) [18] and Brekke and Straume (2004) [19] by assuming that there are two input suppliers, Supplier $A$ and Supplier $B$, of essential inputs. However, differently from Matsushima (2004) [18] and similar to Brekke and Straume (2004) [19], we assume that there is no substitutability between the suppliers. That is, Supplier $A$ ($B$) only

serves Firm $A$ ($B$). Hence, we consider firm-specific input suppliers, rather than competition between the suppliers. We indicate the location of the supplier by $s_J$, $J = A, B$. Like the firms, the suppliers are not constrained to be located within the linear segment of length 1. We depart from Brekke and Straume (2004) [19] by not constraining each supplier to being located in the same place as the firm. In other words, $s_J$ might be different from $x_J$, whereas in Brekke and Straume (2004) [19] $s_J = x_J$. The marginal costs of the suppliers are assumed to be zero for simplicity. We assume that the firms need only the inputs produced by the suppliers. We assume that one unit of input is required to produce one unit of the final goods.

The suppliers carry the input to the firms. Each firm pays the transportation costs of the supplier (for example, it pays the courier carrying the input from $s_J$ to $x_J$). Suppose that the transportation costs from $s_J$ to $x_J$ are also quadratic in the distance (see, for example, Liang and Mai, 2006 [34]), that is $\tau(x_J - s_J)^2$, where $\tau > 0$ is the respective transportation cost parameter for the shipping service from the supplier to the firm.[10] For brevity, we refer to $\tau$ as the transportation cost of the firms.[11] Furthermore, each firm pays the wholesale price to its supplier. We indicate by $w_J$, $J = A, B$, the wholesale price paid to Supplier $J$. We assume for simplicity that besides the input prices and the transportation costs, the firms do not bear any other costs.

Figure 1 illustrates the Hotelling segment with suppliers, for four possible cases:

Case I: Both the suppliers and the firms are within the segment.
Case II: The suppliers are within the segment, whereas the firms are outside the segment.
Case III: The firms are within the segment, whereas the suppliers are outside the segment.
Case IV: Both the suppliers and the firms are outside the segment.

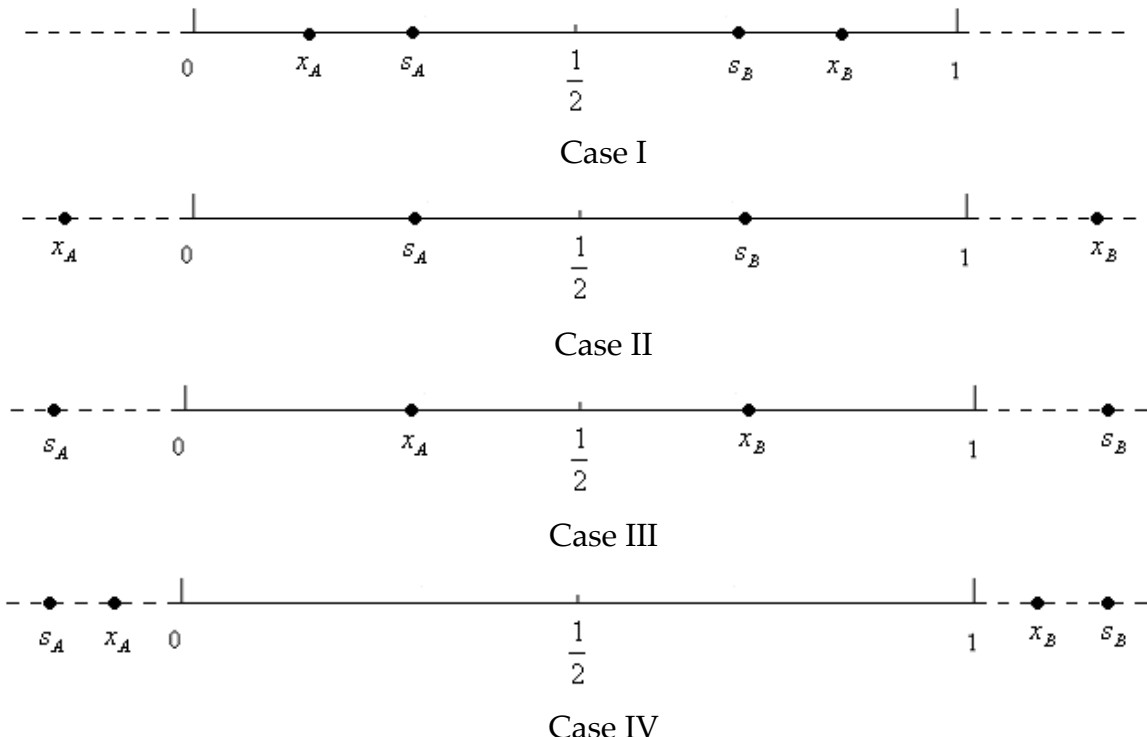

**Figure 1.** The unconstrained Hotelling segment with input suppliers.

## 3. Equilibrium Outcome with Complete Outsourcing

This section considers the basic version of our model where the suppliers' locations are exogenously given, i.e., the suppliers' locations are not influenced by the firms under consideration. In particular, we assume that the suppliers are exogenously and symmetrically located, so that $s_A = s$ and $s_B = 1 - s$, with $s \leq \frac{1}{2}$. Hence, differently from Brekke and

Straume (2004) [19], suppliers might be located in a different place to the firms. Therefore, when $s$ increases, the distance between the suppliers reduces. The analysis of this section helps us to show the implications of the suppliers' locations on the equilibrium outcomes. Later, in Section 5, we consider the case of endogenous locations of the suppliers.

Here, we consider the same three-stage structure and the same non-cooperative nature as in Matsushima (2004) [18], with the exception that the suppliers are firm-specific in our analysis. Therefore, we consider the following game in this section:

Stage 1: The firms decide where to locate simultaneously.

Stage 2: The suppliers decide on the wholesale prices simultaneously.

Stage 3: The firms set the market price simultaneously.

We solve the game by backward induction. The concept adopted for the solution is the subgame perfect Nash equilibrium (SPNE).

We start from the last stage of the game. By solving $v - p_A - t(x - x_A)^2 = v - p_B - t(x - x_B)^2$, we get the consumer which is indifferent between buying from Firm $A$ and from Firm $B$:

$$k = \frac{p_B - p_A + t(x_B^2 - x_A^2)}{2t(x_B - x_A)} \tag{1}$$

Therefore, the demand faced by Firm $A$ is $k$, whereas the demand faced by Firm $B$ is $1 - k$. It follows that, under complete outsourcing, the quantity of input which is purchased from Supplier $A$ by Firm $A$ is $k$, and the quantity of input which is purchased from Supplier $B$ by Firm $B$ is $1 - k$.

Now we consider the profit functions of the firms. As in Matsushima (2004) [18] and Dilek et al. (2018) [37], we assume that the transportation costs endogenously depend on the demand faced by the firms, that is shipping is relative to each unit.[12] Therefore, the profits of Firms $A$ and $B$ are, respectively:[13]

$$\pi_A = [p_A - \tau(s - x_A)^2 - w_A]k \tag{2}$$

$$\pi_B = [p_B - \tau(1 - s - x_B)^2 - w_B](1 - k) \tag{3}$$

By maximizing (2) and (3) with respect to the prices $p_A$ and $p_B$, respectively, we obtain the stage-3 equilibrium prices:

$$p_A = \frac{\tau[1 + 2x_A^2 + x_B^2 - 2x_B(1 - s) - 2s - 4x_A s + 3s^2] - tx_A(2 + x_A) + tx_B(2 + x_B) + 2w_A + w_B}{3} \tag{4}$$

$$p_B = \frac{\tau[2 + x_A^2 + 2x_B^2 - 4x_B(1 - s) - 4s - 2x_A s + 3s^2] - tx_A(4 - x_A) + tx_B(4 - x_B) + w_A + 2w_B}{3} \tag{5}$$

Now, consider stage 2. Under complete outsourcing, the profit functions of Supplier $A$ and Supplier $B$ are, respectively:

$$\pi_{SA} = w_A k \tag{6}$$

$$\pi_{SB} = w_B(1 - k) \tag{7}$$

By maximizing (6) and (7) with respect to the wholesale prices $w_A$ and $w_B$, respectively, we obtain the stage-2 equilibrium wholesale prices as:

$$w_A = \frac{t(x_B - x_A)(8 + x_A + x_B) + \tau(1 - x_A - x_B)(1 + x_A - x_B - 2s)}{3} \tag{8}$$

$$w_B = \frac{t(x_B - x_A)(10 - x_A - x_B) - \tau(1 - x_A - x_B)(1 + x_A - x_B - 2s)}{3} \tag{9}$$

Now we move to the first stage, where the firms decide where to locate. Solving the system composed by the first-order conditions $\frac{\partial \pi_A}{\partial x_A} = 0$ and $\frac{\partial \pi_B}{\partial x_B} = 0$, and imposing firms' symmetry yields a unique solution, namely:[14],[15]

$$x_A* = 1 - x_B* = \frac{4s\tau - 7t}{4(t + \tau)} \tag{10}$$

It is immediate from (10) that when the distance between the suppliers reduces (i.e., $s$ goes up), the distance between the firms reduces as well, as $\frac{\partial x_A*}{\partial s} = \frac{\tau}{t+\tau} > 0$. In addition, the distance between the firms increases with the transportation costs of the consumers, $t$, if the suppliers are close enough (namely, $s > -\frac{7}{4}$), but decreases otherwise.[16] On the contrary, when the transportation costs of the firms, $\tau$, increase, the firms are more separated in equilibrium provided the suppliers are quite distant (namely, $s < -\frac{7}{4}$), but they are closer in equilibrium otherwise.[17]

Note that even if we do not restrict the firms to being located within the segment between 0 and 1, they might locate within the boundaries of the Hotelling segment (i.e., $x_A* > 0$ and $x_B* < 1$). However, it does not happen without strategic input price determination (see, e.g., Lambertini, 1994, 1997 [21,22], and Tabuchi and Thisse, 1995 [23], where the firms always locate outside the segment). In our analysis, this happens when $s > \frac{7t}{4\tau}$, i.e., when the distance between the suppliers and the transportation costs of the consumers are low enough, and/or the transportation costs of the firms are high enough. At the limit, when $s = \frac{1}{2}$ and $\tau \to \infty$ (or $t \to 0$), the two firms agglomerate at the centre of the segment.

Furthermore, it can be observed that $x_A* < s$ (and $x_B* > 1 - s$), and $x_A* \to s$ from the left (and $x_B* \to 1 - s$ from the right) when $\tau \to \infty$ or $t \to 0$ if $s$ is sufficiently high (namely, $s \geq -\frac{7}{4}$). If $s$ is low enough (namely, $s \leq -\frac{7}{4}$), the firms locate within the suppliers, and the location of Firm $A$ ($B$) tends to $s$ ($1 - s$) from the right (left) when $\tau \to \infty$ or $t \to 0$. That is, the firms locate at the same point of the suppliers in the limiting cases where the firms' transportation costs tend to be infinite or there are no consumers' transportation costs.

We summarize the above discussion in the following proposition.

**Proposition 1.** *Under complete outsourcing,*
*(i)    firms locate outside (within) the suppliers when $s \geq (\leq) - \frac{7}{4}$;*
*(ii)   firms locate within the segment between 0 and 1 when $t$ is low enough, and/or $s$ and $\tau$ are high enough;*
*(iii)  the distance between Firms A and B decreases with $s$, increases (decreases) with $t$ if $s \geq (\leq) - \frac{7}{4}$, and increases (decreases) with $\tau$ if $s \leq (\geq) - \frac{7}{4}$.*

In order to explain Proposition 1, it is useful to first compare the equilibrium location in (10) with the results of Brekke and Straume (2004) [19]. In Brekke and Straume (2004) [19], when all the bargaining power belongs to the input supplier, the equilibrium locations are $x_A^* = 1 - x_B^* = -\frac{7}{4}$. Note that this coincides with (10) when there are no transportation costs in the upstream market ($\tau = 0$). Indeed, when the suppliers are located in the same point of the firms, our model is a special case of Brekke and Straume's (2004) [2] framework. Note that this is also true when $s_A^* = 1 - s_B^* = -\frac{7}{4}$, that is, the suppliers locate at firms' locations in equilibrium, or when $s = 1 - s = -\frac{7}{4}$, that is, the suppliers are exogenously located at the firms' locations.

It is well-known that, in a model without price-setting suppliers, the equilibrium location of firms is determined by two contrasting effects: the strategic effect, suggesting that each firm would like to separate from the rival firm in order to soften competition in the final goods market, and the demand effect, suggesting that each firm would like to move closer to the rival firm in order to increase the demand (Tirole, 1988) [38]. Without suppliers, the two effects equate at $x_A^* = 1 - x_B^* = -\frac{1}{4}$ if there are no location constraints (Lambertini,

1994 and 1997 [21,22], and Tabuchi and Thisse, 1995 [34]); in the case of location constraints, the strategic effect dominates, so that the firms maximally differentiate in equilibrium (d'Aspremont et al., 1979) [3].

When adding price-setting input suppliers, a third effect comes up. Even if the two input suppliers do not compete directly, there is indirect competition among them. Indeed, by setting a lower input price, a supplier can induce its downstream firm to reduce its price, which increases the demand for the final goods and therefore, the supply of the input, which, in turn, results in lower sales for the other input supplier, all else equal. The third effect that is introduced by price-setting input suppliers is related to how the locations of the downstream firms affect the (indirect) competition between the input suppliers. In fact, this third effect is the sum of two counteracting sub-effects. If one of the downstream firms locates closer to its rival, this will intensify competition between the two input suppliers and lead to a lower input price and therefore, lower costs for the firm that relocates. However, it will also induce the other supplier to set a lower input price and therefore, lead to lower costs for the competing downstream firm. In the model *á la* Brekke and Straume (2004) [19], the second sub-effect dominates. Hence, the presence of price-setting input suppliers causes the downstream firms to locate even further away from each other ($x_A^* = 1 - x_B^* = -\frac{7}{4}$ instead of $x_A^* = 1 - x_B^* = -\frac{1}{4}$) due to a raising rivals' cost effect: by locating further away from each other, competition between the input suppliers is reduced, leading to higher input prices, which, in turn, dampens competition in the downstream market, leading to higher prices of the final goods as well.[18]

In this paper, we include the possibility that the firms incur transportation costs to procure the inputs they need to produce the final goods. This adds a fourth effect to the analysis, namely, a cost-saving effect: basically, all else equal, each downstream firm tries to save costs by locating closer to its input supplier. This explains the importance of the threshold $s = -\frac{7}{4}$. If $s = -\frac{7}{4}$, there are no transportation costs in equilibrium, so our model collapses into Brekke and Straume's (2004) [19] framework. Therefore, the cost-saving effect vanishes in equilibrium. However, if $s \neq -\frac{7}{4}$, the location choices of the downstream firms are also determined by the aforementioned cost-saving effect, which comes on top of the other effects. Compared with the benchmark of Brekke and Straume (2004) [19], the downstream firms will relocate in the direction of the input suppliers. Thus, if $s < -\frac{7}{4}$, the equilibrium locations are such that the distance between the downstream firms is larger than the benchmark, whereas the opposite is true for $s > -\frac{7}{4}$. And how far away from the benchmark locations the firms will locate depends on the relative strength of the cost-saving effect, which, in turn, depends on the relative magnitudes of $\tau$ and $t$; the cost-saving effect depends on $\tau$ whereas all the other competition-related effects depend on $t$.

Now, we turn to the equilibrium prices and profits. By using (10), we obtain:[19]

$$p* = \frac{t[64\tau^2(1-2s) + t\tau(401 - 72s + 16s^2) + 288t^2]}{16(t+\tau)^2} \tag{11}$$

$$\pi* = \Gamma/4 \tag{12}$$

$$w* = 3\Gamma/2 \tag{13}$$

$$\pi_S* = 3\Gamma/4 \tag{14}$$

where $\Gamma \equiv \frac{t[2\tau(1-2s)+9t]}{(t+\tau)}$. It can be observed that the equilibrium final goods prices and the firms' profits decrease (increase) with the transportation costs of the firms when the suppliers are close (far apart) enough, i.e., $s \geq -\frac{7}{4}$ ($s \leq -\frac{7}{4}$) and increase with the transportation costs of the consumers. When transportation costs of the firms go up and the suppliers are not too distant, we know from Proposition 1 that the firms locate closer to each other. Hence, competition is fiercer in the final goods market, and, consequently, the market prices

and the firms' profits are lower in equilibrium, thus also lowering the wholesale prices and the suppliers' profits. The opposite holds true when the suppliers are distant.

Similarly, when the transportation costs of the consumers increase, all else being equal, the prices of the firms are higher. Even if the firms might end up locating closer in equilibrium (see Proposition 1), when $t$ goes up, the firms' profits increase as well. Analogously, the suppliers' prices and profits also increase with $t$.

Finally, we consider the impact of a lower distance between the suppliers. When the suppliers are located closer to the centre of the market, they are less differentiated. We have also shown that the firms turn out to be less distant in equilibrium. Therefore, the prices and the profits of the suppliers and the firms are lower.

We summarize the above discussions in the next proposition.

**Proposition 2.** *Under complete outsourcing, the equilibrium prices and profits of the firms and the suppliers increase with t, decrease with s, and decrease (increase) with $\tau$ if $s \geq (\leq) - \frac{7}{4}$.*

Now consider the effects on the consumer surplus, which is:[20]

$$CS* = \int_0^{1/2} (v - p* - t(x - x_A*)^2)dx = \frac{v}{2} - \frac{t[1057t + 4\tau(49 - 6s(17 - 2s))]}{96(t + \tau)} \quad (15)$$

Note that $\frac{\partial CS*}{\partial \tau} \geq 0$ when the suppliers are close enough (i.e., $s \geq -\frac{7}{4}$), and $\frac{\partial CS*}{\partial \tau} \leq 0$ otherwise. Therefore, when the suppliers are not too far apart, the consumer surplus unambiguously increases with the transportation costs of the firms. Although it may look puzzling, it happens for two reasons. First, when the suppliers are not too far apart, a higher transportation cost of the firms reduces the wholesale price and the price in the final goods market, as explained above. Second, lower distance between the firms in the final goods market also affects the transportation costs of the consumers. As long as $x_A^* \leq \frac{1}{4}$, the transportation costs of the consumers decrease with $\tau$, but for $x_A^* \geq \frac{1}{4}$, they increase with $\tau$.[21] Therefore, when $x_A^* \leq \frac{1}{4}$, both the effects help to increase the consumer surplus. When $x_A^* \geq \frac{1}{4}$, the first effect dominates the second effect and the consumer surplus increases with the transportation costs of the firms when the suppliers are close enough. The opposite holds true when the suppliers are quite distant ($s \leq -\frac{7}{4}$).

Furthermore, the consumer surplus always decreases with the transportation costs of the consumers. A higher $t$ yields higher prices and increases firms' distance in the final goods market when the suppliers are close to each other (i.e., $s \geq -\frac{7}{4}$). Both the effects push the consumer surplus down. When the suppliers are quite separated (i.e., $s \leq -\frac{7}{4}$), a higher $t$ increases the prices but reduces firms' distance in the final goods market. However, the former effect dominates the latter, and higher transportation costs of the consumers reduce consumer surplus.

Finally, the impact of a lower distance between the suppliers increases consumer surplus. When the suppliers are closer to each other, firms' distance in the final goods market and the prices of the final goods reduce. The price effect dominates the distance effect and increases consumer surplus when the distance between the suppliers is reduced.

The above discussion gives the following result.

**Proposition 3.** *Under complete outsourcing, the consumer surplus decreases with t, increases with s, and increases (decreases) with $\tau$ if $s \geq (\leq) - \frac{7}{4}$.*

Now we briefly discuss the effects on welfare, which is $W* = CS* + \pi* + \pi_S*$. Due to the assumption of inelastic demand functions, the welfare inversely depends only on the total transportation costs, which are determined by both the consumers' transportation costs and the firms' transportation costs. That is, $W* = \frac{v}{2} - TC^c - TC^f$, where $TC^c \equiv \int_0^{1/2} t(x - x_A*)^2 dx$ and $TC^f \equiv \int_0^{1/2} \tau(s - x_A*)^2 dx$. It can be observed that the first-best locations consist of the two firms locating at $1/4$ and $3/4$, respectively. These

locations minimize the consumers' transportation costs, $TC^c$, and each supplier locates at the same point of the firms to minimize the firms' transportation costs, $TC^f$.

First, consider the impact of $t$. We obtain $\frac{\partial W*}{\partial t} = -\frac{193t^2 + 386t\tau + 4\tau^2(1 - 6s + 12s^2)}{96(t+\tau)^2} \leq 0$.
When $t$ increases, for given locations, the consumers' transportation costs increase too, thus lowering welfare. Furthermore, the greater is $t$, the lower the incentive is for the firms to locate closer to the suppliers (see the discussion about Proposition 1), thus increasing the firms' transportation costs and contributing to lower welfare. Even if a greater $t$ might induce the firms to locate closer to the first and the third quartile (see Proposition 1 (iii)), this is not enough to outweigh the negative effects of higher consumers' transportation costs. Hence, welfare decreases with $t$.

Now consider the impact of $\tau$. We observe $\frac{\partial W*}{\partial \tau} = \frac{t^2(63 + 8s - 16s^2)}{32(t+\tau)^2} \leq (\geq) \, 0$ if $s \leq (\geq) - \frac{7}{4}$.
When $\tau$ increases, for given firms' locations, the firms' transportation costs increase as well, which tends to lower welfare. However, when $\tau$ is high, the firms tend to locate closer to the suppliers, and this, all else being equal, reduces the firms' transportation costs, which creates a positive impact on welfare. Furthermore, if $s$ is close to the first quartile, a higher $\tau$ also contributes to lower the consumers' transportation costs. Hence, when $s$ is sufficiently high (namely, $s \geq -\frac{7}{4}$), welfare increases with $\tau$; otherwise, welfare decreases with $s$.

Finally, consider the impact of $s$. We observe $\frac{\partial W*}{\partial s} = \frac{t\tau(1 - 4s)}{4(t+\tau)} \leq (\geq) \, 0$ if $s \geq (\leq) \frac{1}{4}$.
Ceteris paribus, the existence of firm-specific suppliers creates an incentive for the firms to locate closer to them (due to the cost-saving effect). Therefore, as long as the suppliers locate closer to the first and the third quartile, the consumers' transportation costs decrease and welfare increases.

The following proposition follows from the above discussion.

**Proposition 4.** *Under complete outsourcing, welfare decreases with t, increases (decreases) with $\tau$ if $s \geq (\leq) - \frac{7}{4}$, and increases (decreases) with s if $s \leq (\geq) \frac{1}{4}$.*

## 4. Equilibrium Outcome with Bi-Sourcing

In this section, we introduce bi-sourcing into the above framework. That is, each firm might decide how much to produce in-house and how much to purchase from the suppliers. In particular, as in Beladi and Mukherjee (2012) [31], we assume that each firm might build up a capacity for input production to produce $q_J$ units of the final goods in-house and hire workers accordingly for in-house input production; it will produce up to this point, as it has already incurred the costs for input production. However, if the firm needs to produce more than $q_J$ units of the final goods and, therefore, needs more inputs, it buys the extra inputs from the suppliers by paying the wholesale price, $w_J$. The firms determine the in-house capacity and the total final goods production endogenously. In particular, we assume that in-house production is costless. Interestingly, even if there are no costs to produce in-house, the downstream firms might find it convenient to outsource part of the production to external suppliers (see later).

The timing of the game follows Beladi and Mukherjee (2012) [1], with the addition of the first-stage locational choice as in Matsushima (2004) [18] and Brekke and Straume (2004) [19]. In other words, we introduce an intermediate stage with respect to the model discussed in Section 3, where the two firms decide—simultaneously and independently—how much to produce in house.

Now the timing of the game becomes:

Stage 1: The firms decide where to locate simultaneously.
Stage 2: The firms decide how much to produce in-house simultaneously.
Stage 3: The suppliers decide on the input prices simultaneously.
Stage 4: The firms set the prices of the final goods simultaneously.
As usual, we solve the game through backward induction.

The indifferent consumer in the last stage of the game is given by (1). By recalling that $k$ $(1 - k)$ is the demand of Firm $A$ ($B$), the profit functions of the firms are as follows:[22]

$$\pi_A = [p_A - \tau(s - x_A)^2 - w_A](k - q_A) + p_A q_A \tag{16}$$

$$\pi_B = [p_B - \tau(1 - s - x_B)^2 - w_B](1 - k - q_B) + p_B q_B \tag{17}$$

Note that when $q_J = 0$, we are back to the case of complete outsourcing, described in Section 3. Because the overall demand faced by Firm $A$ ($B$) is $k$ $(1 - k)$, $k - q_A$ $(1 - k - q_B)$ is the extra quantity that Firm $A$ ($B$) needs outsourcing to Supplier $A$ ($B$). Therefore, the first term in (16) and (17) indicates the profit arising from the quantity which is not produced in-house, and the second term indicates the profit arising from the quantity that is produced in-house.

Clearly, the last stage equilibrium prices are given by (4) and (5). Now, we consider stage 3. The profit functions of Supplier $A$ and Supplier $B$ are, respectively:

$$\pi_{SA} = w_A(k - q_A) \tag{18}$$

$$\pi_{SB} = w_B(1 - k - q_B) \tag{19}$$

By maximizing, we get the stage-3 equilibrium wholesale prices:

$$w_A = \frac{t(x_B - x_A)(8 + x_A + x_B - 6q_B - 12q_A) + \tau(1 - x_A - x_B)(1 + x_A - x_B - 2s)}{3} \tag{20}$$

$$w_B = \frac{t(x_B - x_A)(10 - x_A - x_B - 12q_B - 6q_A) - \tau(1 - x_A - x_B)(1 + x_A - x_B - 2s)}{3} \tag{21}$$

Now consider stage 2, where Firms $A$ and $B$ choose the in-house production level. The stage-2 equilibrium in-house quantities are:

$$q_A = \frac{\tau[22 + 47x_A^2 + 22x_B^2 - 44x_B(1 - s) - 44s - 94x_A s + 69s^2)] + 11t(x_B - x_A)[18 + 5(x_B + x_A)]}{690t(x_B - x_A)} \tag{22}$$

$$q_B = \frac{\tau[47 + 22x_A^2 + 47x_B^2 - 94x_B(1 - s) - 94s - 44x_A s + 69s^2] + 11t(x_B - x_A)[28 - 5(x_B + x_A)]}{690t(x_B - x_A)} \tag{23}$$

Because we have the equilibrium values for stages 2–4, we can use them to solve stage 1. Because the expressions for the equilibrium locations of the firms are extremely long, we illustrate them for some specific locations of the suppliers.[23] Figure 2 illustrates the equilibrium location of Firm $A$, the in-house production level, and the firms' profits. Because the equilibrium prices and the suppliers' profits follow the same pattern of the firms' profits, we do not report them. The equilibrium values under bi-sourcing are indicated by the superscript "^" to distinguish them from the case of complete outsourcing.

Because $1/2$ $(=k)$ is the equilibrium demand of each firm, bi-sourcing emerges in equilibrium if $0 < \hat{q}* < 1/2$. This requires that $\tau$ is sufficiently low (see the diagrams), which is assumed from now onwards. Interestingly, there are conditions such that the downstream firms prefer to outsource part of the production even if they are more efficient than the input suppliers (recall that in-house production is costless). Indeed, the existence of input suppliers dampens competition in the downstream market: when comparing a situation with input suppliers and one without input suppliers, all else equal, the presence of input suppliers induce the downstream firms to locate further away from each other (this is the raising rival's cost effect discussed in Section 3 and introduced by Brekke and Straume, 2004) [19].

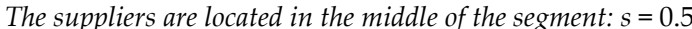

*The suppliers are located in the middle of the segment: s = 0.5*

*The suppliers are located at the endpoints of the segment: s = 0*

*The suppliers are located outside the segment: s = −1*

**Figure 2.** Bi-sourcing equilibrium.

Now consider the impact of the model's parameters on the equilibrium outcomes under bi-sourcing. It can be observed from Figure 2 that higher transportation costs of the firms, $\tau$, increase the incentive to produce in-house (indeed, $\hat{q}*$ always increases with $\tau$). Because the firms produce more in-house, it becomes less important for them to locate closer to the suppliers in order to save on the transportation costs. In other words, the cost-saving effect is less important under bi-sourcing. So, we observe that the equilibrium location of Firm $A$, $\hat{x}_A*$, is more distant from the supplier when $\tau$ increases. This implies that, when Supplier $A$ is located at $1/2$ and $0$ (upper and intermediate picture, respectively), the firms' distance increases with $\tau$, and the profits of the firms increase. On the other hand, when Supplier $A$ is located at $-1$ (lower picture) the firms' distance decreases with $\tau$, and the profits decrease. Therefore, the different impact of $\tau$ on the equilibrium distance is driven by the location of the suppliers. When the suppliers are close, higher transportation costs of the firms push them far apart, but when the suppliers are distant, higher transportation costs of the firms push them closer in space.

Now, we briefly discuss the impacts of $t$. As explained in Section 3, the impact of $t$ on the equilibrium distance of the firms is opposite to that of $\tau$. Furthermore, the higher $t$ is, the lower the quantity produced in-house is, because the cost-saving effect becomes relatively less important when $t$ goes up, which decreases the incentive to produce in-house. Finally, as standard in the Hotelling models, when the transportation costs of the consumers go up, the competition is less fierce and the profits are greater.

Figure 2 can be used to observe the implications of a longer distance between the suppliers. Indeed, all else being equal, lower $s$ implies smaller distance between the firms and creates lower profits. This is due to the fact that, by distancing from the supplier, each firm commits to increase in-house production in the second stage of the game. This result sharply contrasts with Proposition 1(iii), which is for the case of complete outsourcing.

We summarize the impacts of $s$, $t$, and $\tau$, in the next proposition:

**Proposition 5.** *Under bi-sourcing,*

(i)    *the distance between Firms A and B increases with s, decreases (increases) with t if s is high (low), and increases (decreases) with $\tau$ if s is high (low).*

(ii)    *the equilibrium prices and profits of the firms and the suppliers increase with t, increase with s, and increase (decrease) with $\tau$ if s is high (low).*

When comparing Proposition 5(*i*) with Proposition 1(*iii*) (complete outsourcing), it is immediate that bi-sourcing dramatically changes the sign of the impact of the relevant parameters ($s$, $t$, and $\tau$) on the equilibrium distance between firms. Similarly, when comparing Proposition 5(*ii*) with Proposition 2, the impacts of $s$ and $\tau$ on the equilibrium prices and profits under complete outsourcing are opposite to those under bi-sourcing.[24]

Now, we compare the profits under bi-sourcing with the profits under complete outsourcing, in order to illustrate the role played by bi-sourcing in Matsushima (2004) [18]. As discussed above, all else being equal, bi-sourcing reduces the importance of the cost-saving effect, because part of the inputs is procured in-house. Therefore, when the suppliers are close, the incentive for firms to locate closer to the centre is stronger under complete outsourcing than under bi-sourcing. On the contrary, when the suppliers are far away, the incentive for firms to locate closer is stronger under bi-sourcing than under complete outsourcing. In other words, when $s$ is high (low), bi-sourcing works as a centrifugal (centripetal) force, and the firms' distance and profits are higher (lower) under bi-sourcing compared to complete outsourcing.

When considering the transportation costs under complete outsourcing, an increase in $\tau$ ($t$) increases (decreases) the cost-saving effect, whereas under bi-sourcing it reduces (increases) the cost-saving effect via the expansion (reduction) of in-house production. Therefore, if $s$ is high enough, the firms' distance and profits are more likely to be higher under bi-sourcing compared to complete outsourcing when the firms' (consumers') transportation costs are high (low).

Figure 3 illustrates the above discussion, and Proposition 6 shows the implications of bi-sourcing:[25]

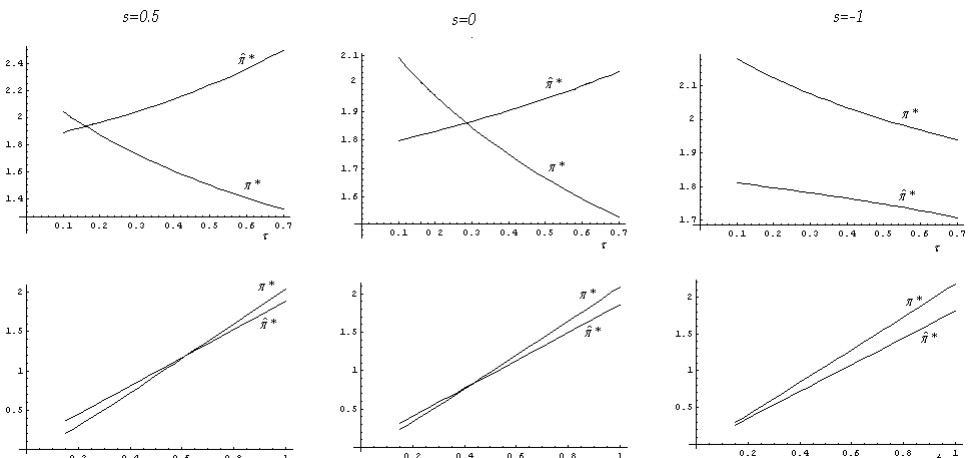

**Figure 3.** Equilibrium profits under bi-sourcing and complete outsourcing.

**Proposition 6.** *Firms' distance and profits are higher under bi-sourcing compared to complete outsourcing when s and τ are high enough, and t is low enough.*

We conclude this section by discussing the case of bi-sourcing under competition between the suppliers instead of firm-specific suppliers, so that we consider bi-sourcing in a model that is otherwise similar to Matsushima (2004) [18]. The last stage equilibrium prices are given by Equations (18) and (19). Now we consider stage 3. Following Matsushima (2004, Equations (6) and (7)) [25] and a standard Bertrand argument, the equilibrium prices set by the suppliers are such that each supplier fixes the wholesale price at the rival's transportation cost (provided that its cost is lower than that of the rival). Therefore, in equilibrium, Supplier $A$ ($B$) serves Firm $A$ ($B$) by setting $w_A = \tau(1 - s - x_A)^2$ ($w_B = \tau(s - x_B)^2$). By using these prices in the downstream firms' profits (Equations (16) and (18)), we observe that the profits are strictly increasing in the in-house production level. Indeed, $\frac{\partial \pi_J}{\partial q_J} = \tau(1 - 2x_J + 2x_J^2 - 2s + 2s^2) > 0$, $J = A, B$, that is, each firm decides to procure all the inputs in-house, implying no bi-sourcing in equilibrium.[26] Therefore, bi-sourcing will not occur in equilibrium in Matsushima (2004) [18] and firm-specific input suppliers are necessary to observe bi-sourcing in equilibrium.[27,28]

## 5. Extensions

In this section, we extend the model discussed so far in two directions: (i) the suppliers are located at the same place as the firms as in Brekke and Straume (2004) [19], and (ii) the suppliers choose where to locate. The analysis of this section will show that the maximum differentiation result of d'Aspremont et al. (1979) [3] comes back in these cases. Hence, whether the maximum differentiation result of d'Aspremont et al. (1979) [3] remains in the presence of strategic input price determination may depend on the location choice of the suppliers, i.e., whether the locations of the suppliers are not influenced by the firms under consideration (the case of exogenous locations of the suppliers) or whether the suppliers locate at the same place of the firms or the suppliers' locations are influenced by the firms under consideration (endogenous locations of the suppliers).

### 5.1. The Suppliers Are Located at the Same Place of the Firms

In this extension, we assume that Supplier $A$ ($B$) is located at the same point as Firm $A$ ($B$).[29] Note that this case is immediately comparable to Brekke and Straume's (2004) [19] framework, once all the bargaining power belongs to the input suppliers.

If the model runs as in Section 4, once we set $s = x_A$ and $1 - s = x_B$, we obtain the equilibrium outcomes as: $x_A* = 1 - x_B* = -\frac{13}{20}$, $p* = \frac{207t}{50}$, $q* = \frac{11}{30}$, $\pi* = \frac{2737t}{1500}$, $w* = \frac{46t}{25}$, and $\pi_S* = \frac{92t}{375}$.

It is interesting to compare this outcome with the case of no bi-sourcing. We find from Equations (13) and (14) of Brekke and Straume (2004) [19] with the full bargaining power of the input suppliers that, in the absence of bi-sourcing, the firms locate at $x_A^{bs} = 1 - x_B^{bs} = -\frac{7}{4}$. Comparing these location choices with our results of $x_A* = 1 - x_B* = -\frac{13}{20}$, it is immediate that bi-sourcing reduces the incentive to locate far apart. The reason is similar to the discussion in Section 4. Because the firms are closer, the profits are lower.[30]

*5.2. The Suppliers Choose Where to Locate*

In this extension, we assume that the suppliers choose endogenously where to locate, so that Supplier $J = A, B$ chooses location $s_J$. Furthermore, we assume that the suppliers and the firms choose their locations simultaneously.[31] The game has the following four stages:

Stage 1: The firms and the suppliers decide where to locate simultaneously.

Stage 2: The firms decide how much to produce in-house simultaneously.

Stage 3: The input suppliers decide on the wholesale price simultaneously.

Stage 4: The downstream firms set the market price simultaneously.

The model runs as in Section 4, and we set $s = s_A$ and $1 - s = s_B$. The analyses of stages 2–4 are the same as in Section 4. When considering stage 1, it is easy to show that each supplier maximizes its profits by locating in the same place as the firms, that is $s_J* = x_J*$.[32] Therefore, we have the same results outlined in Section 5.1. Indeed, each supplier, by locating as close as possible to its respective firm, reduces the transportation costs of the firm and maximizes its own profits.

One might also consider the case of a simultaneous choice of locations by the suppliers and the firms under complete outsourcing. This amounts to considering the game developed in Section 3 with the difference that in the first stage the suppliers also choose their locations. By solving the game, we obtain $x_A* = 1 - x_B* = -7/4$ and $s_J* = x_J*$. Therefore, as for the case of bi-sourcing, the transportation costs of the firms are nullified, and the cost-saving effect vanishes because the suppliers locate at the same point of the firms: hence, we are back to the case discussed in Brekke and Straume (2004) [19]. However, the firms' distance is greater under complete outsourcing compared to bi-sourcing. Indeed, in the Beladi and Mukherjee (2012) [31] framework, the firms commit to in-house production before setting the downstream price. Therefore, in the absence of the cost-saving effect, the incentive to avoid fierce competition is weaker under bi-sourcing. It follows that when both the suppliers and the firms choose their locations, bi-sourcing lowers firms' distance in equilibrium.

## 6. Conclusions

The Hotelling model (1929) [1] represents one of the most relevant game-theoretic applications in economics. By using the concepts of game theory, the extant literature has extensively adopted the Hotelling model to analyze the location decisions of competing firms, and the resulting equilibrium in terms of spatial separation, prices, and profits. However, the game-theoretic applications of the Hotelling model are rather silent about the issue of input procurement by the firms competing in the Hotelling world. Our paper is a step towards filling this gap.

In particular, this paper considers the location choice of the final goods producers in a Hotelling duopoly with firm-specific input suppliers. It contributes to the literature in two ways. First, it extends the line of research of Matsushima (2004) [18] and Brekke and Straume (2004) [19] by showing the effects of the transportation costs of consumers, the transportation costs between the final goods producers and the input suppliers, and the distance between the input suppliers and the location choice of the final goods producers. In this respect, it shows the implications of exogenous and endogenous locations of the input

suppliers. Second, it shows the implications of bi-sourcing by the final goods producers compared to complete outsourcing.

Under complete outsourcing, the final goods producers locate closer as the distance between the input suppliers decreases, but the distance between the final goods producers may increase or decrease with the transportation costs of the consumers and the transportation costs between the input suppliers and the final goods producers depending on the distance between the input suppliers.

In the case of complete outsourcing, a cost-saving effect emerges, as each downstream firm would like to minimize its own transportation cost for procuring the inputs from the external supplier. This leads each downstream firm to locate closer to its supplier. This incentive adds to the traditional demand and strategic effect (Tirole, 1988) [38], as well as to the raising rivals' cost effect (Brekke and Straume, 2004) [19], when determining the equilibrium locations of the downstream firms.

When considering bi-sourcing, we show that the final goods producers' distance and profits might be higher in equilibrium. The benefit from saving the transportation costs between the input suppliers and the final goods producers is lower under bi-sourcing and creates effects which are opposite to those under complete outsourcing. All else being equal, the final goods producers procure less inputs form the suppliers. Therefore, the incentive to locate closer to the input suppliers is low. It follows that when the input suppliers are close to the centre, the incentive to move far from the input suppliers under bi-sourcing is greater than under complete outsourcing, which increases distance between the final goods producers. In this case, bi-sourcing yields higher profits in equilibrium.

Although our analysis provides new insights by considering exogenous and endogenous locations of the input suppliers, and bi-sourcing, several questions remain unexplored and deserve further research. We briefly mention some possible extensions which are worthy of investigation. First, in our model there is no bargaining between each downstream firm and its specific supplier. However, under firm-specific suppliers, it might be natural to consider some bargaining processes (Brekke and Straume, 2004 [19], Du et al., 2006 [32], Feng and Lu, 2013 [39]). It is likely that the locational incentives would be affected by the bargaining power of the downstream firm vs. that of the upstream supplier. Second, we assume the existence of two suppliers, one for each firm (bilateral monopolies). However, one might imagine the existence of a unique supplier serving both downstream firms (Wu et al., 2012) [40]. Because the existence of a monopolistic supplier alters the bi-sourcing outcome in a non-spatial framework (Beladi and Mukherjee, 2012 [31], Colombo and Scrimitore, 2018) [33], we might also expect some implications for the locational equilibrium in a spatial set-up. Finally, we do not explicitly analyze the case of a vertical merger between the downstream firm and its specific supplier (Matsushima, 2004 and 2009) [25,26]. It might be easily shown that in this case both firms would choose complete in-house production, and the locational equilibrium would replicate the one in Lambertini (1994, 1997) [21,22] and Tabuchi and Thisse (1995) [23]. However, it remains to be investigated whether or not vertical integration emerges as an equilibrium when downstream firms and upstream suppliers are required to choose their integration policy before the game starts. We leave these questions for further research.

**Author Contributions:** Conceptualization, S.C. and A.M.; Formal analysis, S.C. and A.M.; Writing—original draft, S.C. and A.M.; Writing—review & editing, S.C. and A.M. All authors have read and agreed to the published version of the manuscript.

**Funding:** This research received no external funding.

**Data Availability Statement:** Not applicable.

**Acknowledgments:** We thank three anonymous referees of this journal. We also thank the participants of the 2023 Taipei Conference in Regional Science, the SIEPI 2023 Conference in Naples (Italy), the SIE 2023 Conference in L'Aquila (Italy) and, in particular, Fu-Chuan Lai and Wen-Jung Liang for their suggestions. Usual disclaimers applies.

**Conflicts of Interest:** The authors declare no conflict of interest.

## Notes

[1]   Ebina et al. (2015) [10] take into consideration the product positioning of two firms in a spatial competition model with a continuous time and sequential entries. Ebina et al. (2022) [11] extend their previous analysis by incorporating demand uncertainty.

[2]   Interestingly, input procurement by a spatially localized firm has received some attention in a different tradition. Indeed, the literature adopting the Weber triangle (Weber, 1909) [12] focuses on the problem of a monopolistic firm localized in a point aiming to procure inputs (located in other places) in order to produce goods to be sold in a market at a certain distance from the firm's plant (see for instance Moses, 1958 [13], Sakashita, 1967 [14], Shieh and Mai, 1997 [15], Tan, 2001) [16]. Therefore, there are transportation costs to procure the inputs and transportation costs to sell the goods. However, as clearly pointed by Lai and Tabuchi (2012) [17], "Weber (1909) [12] is more realistic in terms of the fact that manufacturing firms use inputs in producing a final product. However, Hotelling (1929) [14] is more realistic in terms of the fact that there is competition between firms" (p. 1017). In this paper, we incorporate the issue of input procurement in the Hotelling model because we want to analyze its implications for firms competing both in locations and prices.

[3]   For example, asset specificity might be due to irreversible R&D expenditures or sunk marketing expenditures that, by increasing the value of trade between the downstream firm and the input supplier, generate a switching cost that might preclude any outside option for both the firm and the supplier. Joskow (1991) [20] provides an empirical survey that widely documents the existence of firm-specific input suppliers.

[4]   For example, Sony internally produces display panels—which are an input for the final goods (namely, LCD TVs)—but it also procures display panels from professional panel suppliers such as AU Optronics (see Lin et al., 2016) [24].

[5]   This situation is better suited to describe the relationship between the labor force ("input" supplier) and the downstream firm. By contrast, our framework, where the suppliers might be far apart from the firms, describes those cases where the input supplier is a firm too.

[6]   Matsushima and Mizuno (2012a) [27] consider a similar setting with more than four players by allowing for asymmetry between the players. Matsushima and Mizuno (2012b) [28] instead develop a non-spatial framework where each downstream firm requires two (complementary) inputs rather than one.

[7]   Lai and Tabuchi (2012) [17] also consider input procurement within the context of a traditional location-price Hotelling game with quadratic transportation costs. However, in that paper, the inputs are simply raw materials which are located in space and are freely available to the firms. Therefore, there are no input suppliers which set the prices of the inputs.

[8]   Another somehow related paper is by Liang and Mai (2006) [34]. In a spatial competition model, they consider two firms, and one of them could subcontract the production of inputs to the (more efficient) rival. Therefore, differently from our paper, procurement of input is not from an upstream supplier, but from a rival operating at the same market level.

[9]   There is a study analyzing equilibrium product differentiation in non-spatial models. See Han et al. (2022) [35] and the references therein. However, that study neither shows the effects of the distance between the input suppliers and different types of transportation costs nor considers the effects of bi-sourcing.

[10]  Alternatively, one might assume that each supplier incurs its own transportation costs. The results would be qualitatively the same.

[11]  Following a consolidated tradition in the Hotelling models, we consider both the consumers' transportation cost, $t$, and the firms' transportation costs, $\tau$, as exogenous variables. In a recent paper, Kucera and Kaderabkova (2023) [36] treat the transportation costs as an endogenous variable. However, they do not consider the problem in hand.

[12]  Alternatively, the transportation costs might not depend on the demand. This case refers to a situation in which what matters is the freight cost per se, for instance that of a truck, irrespective of its load.

[13]  Under the alternative specification of the transportation costs, where transportation costs do not depend on the demand, the profit functions would be $\pi_A = (p_A - w_A)k - \tau(s - x_A)^2$ and $\pi_B = (p_B - w_B)(1 - k) - \tau(1 - s - x_B)^2$. It can be shown that there is no qualitative difference in the results under the two specifications. Hence, we only focus on the case where the transportation costs depend on the demand.

[14]  It can be observed that the second-order conditions are satisfied at (10).

[15]  When the transportation costs of the firms do not depend on the demand, the equilibrium locations could be found as $\tilde{x}_{A}* = 1 - \tilde{x}_{B}* = \frac{36\tau s - 7t}{4(9\tau + t)}$. Therefore, the results are qualitatively similar to those emerging from the model discussed in the text.

[16]  We get $\frac{\partial x_A*}{\partial t} = -\frac{\tau(7 + 4s)}{4(t + \tau)^2}$.

[17]  We get $\frac{\partial x_A*}{\partial \tau} = \frac{t(7 + 4s)}{4(t + \tau)^2}$.

[18]  In a model where total demand is inelastic, higher input prices can be easily passed on to the consumers in the form of higher output prices, implying that the downstream firms benefit from higher input prices in a symmetric equilibrium.

[19]  Due to symmetry, we omit the subscripts when not necessary.

20    Due to symmetry, we consider just the left-hand side of the market.

21    The transportation costs of the consumers are minimized when $x_A* = 1/4$.

22    As for Section 3, the results in this section would be similar by assuming that the transportation costs of the firms do not depend on the demand. Details are available upon request.

23    The equations for the firms' equilibrium locations are available upon request.

24    Only the impact of *t* is the same (and positive), as usual in the Hotelling models.

25    Due to the complexity of the equations involved, the proof of Proposition 6 has been performed by using Mathematica software (Mathematica, Version 5) and it is available on request. The upper (lower) graphs in Figure 3 are drawn for $t = 1$ ($\tau = 1/10$).

26    Note that the complete in-house production result depends on our assumption of zero production costs of the downstream firms. However, it can be shown that introducing positive production costs of the firms does not affect the linearity of firms' profits with respect to the in-house quantity level. In particular, if the production costs are sufficiently low, complete in-house production emerges in equilibrium, whereas if they are sufficiently high, complete outsourcing occurs. In any case, bi-sourcing never arises.

27    This result is consistent with previous findings in the bi-sourcing literature. See for instance Beladi and Mukherjee (2012) [31] and Colombo and Scrimitore (2018) [33].

28    We do not explicitly consider consumer surplus and welfare under bi-sourcing. However, because the impacts of *s* and $\tau$ on the firms' distance, prices and profits, are opposite to those under complete outsourcing, it is intuitive that their impacts on consumer surplus and welfare are also opposite. When considering *t*, we know that the impact of *t* on the firms' distance is opposite in the two frameworks. Therefore, the impact on welfare (which depends only on the firms' distance) is likely to be opposite too. However, no clear prediction can be made for the impact of *t* on consumer surplus, which depends on both firms' distance and the equilibrium prices and profits.

29    For example, one might imagine that the firms and suppliers are vertically integrated. Firm *J* is the headquarters of the vertically-integrated firm, whereas supplier *J* is the internal division producing the inputs and charges a transfer price to the headquarters.

30    Indeed, without bi-sourcing, the equilibrium profits of the firms are $9t/4$.

31    The results would be the same if the suppliers and firms choose sequentially, with the suppliers choosing first and the firms choosing second.

32    The details are available upon request.

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
