# Peer review of "Location of Firms and Outsourcing"

_games, doi:10.3390/g14060070_

Round 1

Reviewer 1 Report

Comments and Suggestions for Authors

Thank the authors for this interesting article on the location of the firm producing the final output given the choices of either opting for outsourcing or bi-sourcing based on input prices. Under outsourcing, it is argued that the firms locate closer to the suppliers depending on the transportation cost of consumers and suppliers. Under bi-sourcing the reverse happens. Typically, under bi-sourcing firms use the payoff from one supplier as a backup option in negotiating with the other supplier.

In terms of consumer surplus, any increase in the price leads to a reduction in consumer surplus and vice versa. Transportation cost adds to the price, so firms try to locate their production facilities where costs can be reduced. In the production stage, Firms that produce the final product typically locate their facility close to the suppliers (Producer-Suppliers).

After final production, any cost incurred by the consumer towards transportation for consuming the product will also reduce the consumer surplus. Hence, the firm’s location shall be close to the consumers to reduce transportation costs for them (Producer-Consumers). Lines 71 to 80 are not very clear to the reader, it is better to explain the idea clearly. It is argued that both types of transportation costs (Producer-Suppliers and Producer-consumers) have a similar effect on welfare as both reduce the price for the consumers.

Lines-82-84 the effect of distance between suppliers on welfare differs from that of the consumers. I presume that this is a distance between consumers and final producers. If so please clarify in the text.

Lines 84-88 are unclear on the lower distance between suppliers increases welfare …………….. especially the inverted U-shaped relationship. Distance between suppliers, the distance between suppliers-firms, Welfare. 

The article is interesting and the limitations are well-written which can be useful for future research.

My general comment is to improve the paper's readability for the general public. 

Comments on the Quality of English Language

General readability can be improved. 

Reviewer 2 Report

Comments and Suggestions for Authors

Referee report for: “Location of firms and outsourcing”

This study examines the Hotelling model with quadratic transportation costs (thus, the d’Aspremont e.t.al model) with the added twist of suppliers to the two firms that are also located on the Hotelling line.  Such a situation with such a model has been previously analyzed by Brekke and Straume and Matsushita and Lin and Tu, among others.  How this study differs is that the final goods producers can do bi-sourcing.  The authors find that this possibility of bi-sourcing affects the locations, pricing, profits and also the welfare implications.

I find this type of analysis has ignored a previous literature using the Webberian Triangle (M. J. Weber, 1972, Impact of Uncertainty on Location, MIT Press).  That literature serves as a useful perfect competition benchmark for this study.  The Webberian triangle work also deviates from the assumption used in this and the previous studies where the suppliers as well as the consumers are all located on a line.

The subject of this referee report has been competently done.  It addresses a concern that is of reasonable interest.  Unfortunately, I do not see that the contribution in terms of game theory or with respect to the behavior of firms to be significant enough to merit publication in Games.   

Comments on the Quality of English Language

The quality of english is O.K.

Reviewer 3 Report

Comments and Suggestions for Authors

Dear Authors,

congratulations for your interesting research. Your research is well structured, I found, however, one aspect missing in the article theoretical background. It has been found, recently, that transportation costs may represent an endogenous variable in the Hotelling and Salop models. In this regards, Kucera et al. (DOI 10.52950/ES.2023.12.1.005) Approximate transportation costs of different groups of consumers. I recommend refer to Kucera’s paper including this aspect in the theoretical settlement of your research. Other references are relevant and I am convinced that conclusions reflect research conducted. I may recommend to the authors to express more explicitely the value added  to the current state of art (to underline their novelty) in the conclusion.

Good luck!

Round 2

Reviewer 2 Report

Comments and Suggestions for Authors

The manuscript still does not have results significant enough to merit publication in Games.

Author Response

We are very sorry to see that you are not satisfied with our response to your previous comment. In particular, you say “Unfortunately, I do not see that the contribution in terms of game theory or with respect to the behavior of firms to be significant enough to merit publication in Games.” As a result, you made the further statement that “The manuscript still does not have results significant enough to merit publication in Games.”

Following the suggestion of the Editor, we have now discussed in the Introduction and in the Conclusion how our paper is related to the application of game theory. Please see the yellow highlighted paragraphs in pages 1-2 and 30-31.

By using the concepts of game theory, the extant literature has extensively adopted the Hotelling model to analyse location decisions. Our paper follows that tradition of the literature and uses the solution concepts of game theory to consider the issue of input procurement by the firms competing in the Hotelling world. Since the game-theoretic applications of the Hotelling model is rather silent about the issue of input procurement, our paper is a step to fill this gap.